# The yeast mating-type switching endonuclease HO is a domesticated member of an unorthodox homing genetic element family

Aisling Y Coughlan[1], Lisa Lombardi[1], Stephanie Braun-Galleani[1], Alexandre AR Martos[1], Virginie Galeote[2], Frédéric Bigey[2], Sylvie Dequin[2], Kevin P Byrne[1], Kenneth H Wolfe[1]*

[1]UCD Conway Institute and School of Medicine, University College Dublin, Dublin, Ireland; [2]SPO, INRAE, Université Montpellier, Montpellier SupAgro, Montpellier, France

**Abstract** The mating-type switching endonuclease HO plays a central role in the natural life cycle of *Saccharomyces cerevisiae*, but its evolutionary origin is unknown. *HO* is a recent addition to yeast genomes, present in only a few genera close to *Saccharomyces*. Here we show that *HO* is structurally and phylogenetically related to a family of unorthodox homing genetic elements found in *Torulaspora* and *Lachancea* yeasts. These *WHO* elements home into the aldolase gene *FBA1*, replacing its 3' end each time they integrate. They resemble inteins but they operate by a different mechanism that does not require protein splicing. We show that a WHO protein cleaves *Torulaspora delbrueckii FBA1* efficiently and in an allele-specific manner, leading to DNA repair by gene conversion or NHEJ. The DNA rearrangement steps during *WHO* element homing are very similar to those during mating-type switching, and indicate that *HO* is a domesticated *WHO*-like element.

*For correspondence:
kenneth.wolfe@ucd.ie

Competing interests: The authors declare that no competing interests exist.

## Introduction

Mating-type switching is an important process in the natural life cycle of many budding yeast species. If an uninhabitable environment improves and becomes habitable, any yeast spores that germinate earlier than others will have a competitive advantage, provided that they have a backup mechanism to prevent death if they germinate too early. Mating-type switching provides this backup (*Hanson and Wolfe, 2017*). Yeast spores are survival capsules, and mating-type switching enables the microcolony formed from a newly-germinated spore to sporulate again after just a few cell divisions if necessary, preventing its extinction (*Herskowitz, 1988*). Across the phylogenetic tree of budding yeasts, mating-type switching has arisen independently at least 11 times, indicating strong natural selection in favor of switching (*Krassowski et al., 2019*).

In *Saccharomyces cerevisiae*, *HO* is the central gene in the mating-type switching process. It was one of the first yeast genes ever discovered, because haploid strains with a functional *HO* gene can switch their mating type and hence auto-diploidize and form visible spores, whereas *ho* mutants cannot (*Winge and Roberts, 1949*; *Oshima, 1993*). *HO* codes for an endonuclease that makes a double-strand DNA break at the mating type (*MAT*) locus, and is essential for efficient switching (*Kostriken et al., 1983*; *Russell et al., 1986*). Much of our knowledge about how eukaryotes repair double-strand DNA breaks in their chromosomes comes from studies that used HO as a model system (*Haber, 2016*). But despite the comprehensive functional and genetic characterization of *HO*, its evolutionary origin remains mysterious (*Keeling and Roger, 1995*; *Haber and Wolfe, 2005*;

**eLife digest** In the same way as a sperm from a male and an egg from a female join together to form an embryo in most animals, yeast cells have two sexes that coordinate how they reproduce. These are called "mating types" and, rather than male or female, an individual yeast cell can either be mating type "a" or "alpha". Every yeast cell contains the genes for both mating types, and each cell's mating type is determined by which of those genes it has active.

Only one mating type gene can be 'on' at a time, but some yeast species can swap mating type on demand by switching the corresponding genes 'on' or 'off'. This switch is unusual. Rather than simply activate one of the genes it already has, the yeast cell keeps an inactive version of each mating type gene tucked away, makes a copy of the gene it wants to be active and pastes that copy into a different location in its genome. To do all of this yeast need another gene called *HO*. This gene codes for an enzyme that cuts the DNA at the location of the active mating type gene. This makes an opening that allows the cell to replace the 'a' gene with the 'alpha' gene, or vice versa. This system allows yeast cells to continue mating even if all the cells in a colony start off as the same mating type. But, cutting into the DNA is risky, and can damage the health of the cell. So, why did yeast cells evolve a system that could cause them harm?

To find out where the *HO* gene came from, Coughlan et al. searched through all the available genomes from yeast species for other genes with similar sequences and identified a cluster which they nicknamed "weird *HO*" genes, or *WHO* genes for short. Testing these genes revealed that they also code for enzymes that make cuts in the yeast genome, but the way the cell repairs the cuts is different. The *WHO* genes are jumping genes. When the enzyme encoded by a *WHO* gene makes a cut in the genome, the yeast cell copies the gene into the gap, allowing the gene to 'jump' from one part of the genome to another. It is possible that this was the starting point for the evolution of the *HO* gene. Changes to a WHO gene could have allowed it to cut into the mating type region of the yeast genome, giving the yeast an opportunity to 'domesticate' it. Over time, the yeast cell stopped the WHO gene from jumping into the gap and started using the cut to change its mating type.

Understanding how cells adapt genes for different purposes is a key question in evolutionary biology. There are many other examples of domesticated jumping genes in other organisms, including in the human immune system. Understanding the evolution of *HO* not only sheds light on how yeast mating type switching evolved, but on how other species might harness and adapt their genes.

---

*Koufopanou and Burt, 2005*; *Muller et al., 2007*). The *HO* gene is a relatively recent evolutionary addition into the yeast genome, because it is found only in a few genera closely related to *Saccharomyces* (*Butler et al., 2004*; *Hanson and Wolfe, 2017*). The 'three-locus' system for mating-type switching, involving an active *MAT* locus and silent *HML* and *HMR* loci, pre-dates the origin of *HO*. An outgroup species, *Kluyveromyces lactis*, also has a three-locus system but has no *HO* gene, and employs alternative mechanisms to create a double-strand break at the *MAT* locus to initiate switching (*Barsoum et al., 2010*; *Rajaei et al., 2014*). Some more distantly related budding yeasts switch mating types using 'two-locus' flip/flop inversion systems, and again do not have an *HO* gene (*Hanson and Wolfe, 2017*; *Krassowski et al., 2019*).

As well as being a recent evolutionary innovation, HO endonuclease also has an unusual protein domain structure that begs the question of where it came from. It resembles inteins, but is not an intein itself. Inteins are mobile genetic elements that are completely protein-coding and occur as in-frame fusions within a host gene (*Novikova et al., 2014*). After the host gene is transcribed and translated, the intein is excised post-translationally and the host protein is assembled by protein splicing, making a peptide bond between its N- and C-terminal parts (exteins). HO has highest sequence similarity to the VDE intein of budding yeasts, which is the only intein in *S. cerevisiae* (*Koufopanou and Burt, 2005*; *Green et al., 2018*). The host gene for *VDE* is *VMA1*, which codes for a subunit of vacuolar H$^+$-ATPase (*Gimble and Thorner, 1992*; *Anraku et al., 2005*). The excised VDE intein has endonuclease activity and can cleave empty (inteinless) alleles of *VMA1*, enabling the intein to spread through the population by homing – a selfish, super-Mendelian mode of inheritance

(*Burt and Koufopanou, 2004*; *Burt and Trivers, 2006*). The *VMA1* genes of several species in the budding yeast family Saccharomycetaceae are polymorphic for the presence/absence of *VDE*, due to active homing and interspecies spread of the intein (*Koufopanou et al., 2002*; *Okuda et al., 2003*). Homing of *VDE* into empty alleles of *VMA1* occurs during meiosis in diploids that are heterozygotes for intein-containing and empty alleles of *VMA1* (*Gimble and Thorner, 1992*). The VDE protein has two domains (*Moure et al., 2002*): a protein splicing domain that enables the host protein Vma1 to be made, and a homing endonuclease domain that enables the *VDE* DNA sequence to home into empty alleles of *VMA1*.

Most inteins are found in bacteria and archaea, not yeasts (*Poulter et al., 2007*; *Novikova et al., 2014*; *Green et al., 2018*). In phylogenetic and other sequence similarity analyses, the two yeast proteins HO and VDE were found to be each other's closest relatives (*Dalgaard et al., 1997*; *Koufopanou and Burt, 2005*; *Green et al., 2018*). Although HO is related to inteins, and more distantly related to other homing endonucleases in the LAGLIDADG superfamily (*Chevalier and Stoddard, 2001*), it is an independently expressed standalone gene, whereas inteins and other homing endonucleases are self-splicing entities embedded within their host genes (*Belfort et al., 2005*; *Belfort, 2017*). HO does not undergo protein splicing and has no exteins. HO also has a unique zinc finger domain at its C-terminus that is not present in other intein-like proteins or homing endonucleases. In *S. cerevisiae* HO, amino acid residues essential for cleavage of the *MAT* locus are located both in the zinc finger and in the endonuclease domain of the intein-like region (*Meiron et al., 1995*; *Bakhrat et al., 2004*; *Bakhrat et al., 2006*), and the endonuclease has a stringent requirement for zinc ions (*Jin et al., 1997*).

A key feature differentiating HO from true homing endonucleases is that it does not propagate its own DNA sequence – in other words, it does not home. *HO* has become a normal cellular gene, and is an example of a domesticated mobile genetic element (*Volff, 2006*; *Rusche and Rine, 2010*). However, until now it has been unclear what type of mobile element HO originated from. Here, we show that HO is related to a large and diverse family of intein-zinc finger fusion proteins (WHO proteins) that occur mostly in the yeast genus *Torulaspora*. WHO proteins are encoded by a newly discovered homing genetic element, whose genomic organization is different from all other known homing elements, and whose host is the aldolase gene *FBA1*. The similarities between WHO and HO proteins, and between the DNA rearrangement steps that occur during *WHO* element homing and mating-type switching, show how the HO-catalyzed system of mating-type switching originated.

## Results

### *WHO* genes code for a family of intein-zinc finger fusion proteins similar to HO

In the genome sequence of the type strain of *Torulaspora delbrueckii* (CBS1146; *Gordon et al., 2011*), we identified a cluster of five genes (*TDEL0B06670* to *TDEL0B06710*) spanning 14 kb that have sequence similarity to *HO*. We renamed these genes *WHO1* to *WHO5*, for 'weird *HO*' (*Figure 1A*). Two of them are pseudogenes, with a single frameshift in *WHO1* and more extensive damage in *WHO5*. The *WHO* gene cluster is located downstream of the *FBA1* gene encoding fructose-1,6-bisphosphate aldolase, an enzyme that functions bidirectionally in glycolysis and gluconeogenesis (*Schwelberger et al., 1989*). Amino acid sequence identity among the inferred WHO proteins is unusually low for a tandem gene cluster, ranging from 55% (Who2 vs. Who4) down to 24% (Who3 vs. Who4). *T. delbrueckii* also has an *HO* gene (*TDEL0A00850*) elsewhere in its genome, orthologous and syntenic with the *HO* gene of *S. cerevisiae*. The five inferred WHO proteins have only 22–25% identity to *T. delbrueckii* HO (BLASTP *E*-values in the range 1e-6 to 3e-30).

The length and content of the *WHO* gene cluster is polymorphic among natural isolates of *T. delbrueckii*. This species has a primarily haploid (haplontic) life cycle (*Kurtzman, 2011*), so there is only one allele per strain. We found six different allelic *WHO* cluster arrangements among 15 strains we examined, with content ranging from 2 to 9 *WHO* genes and pseudogenes (*Figure 1A*; *Supplementary file 1*). The largest cluster (18 kb) is in strain L09, with three intact genes (*WHO6*, *WHO2*, *WHO3*) and six *WHO* pseudogenes.

As well as *WHO* genes and pseudogenes, some of the *T. delbrueckii* clusters contain one or two duplicated fragments of the 3′ end of *FBA1*, interspersed with *WHO* genes (*Figure 1A*). The *FBA1*

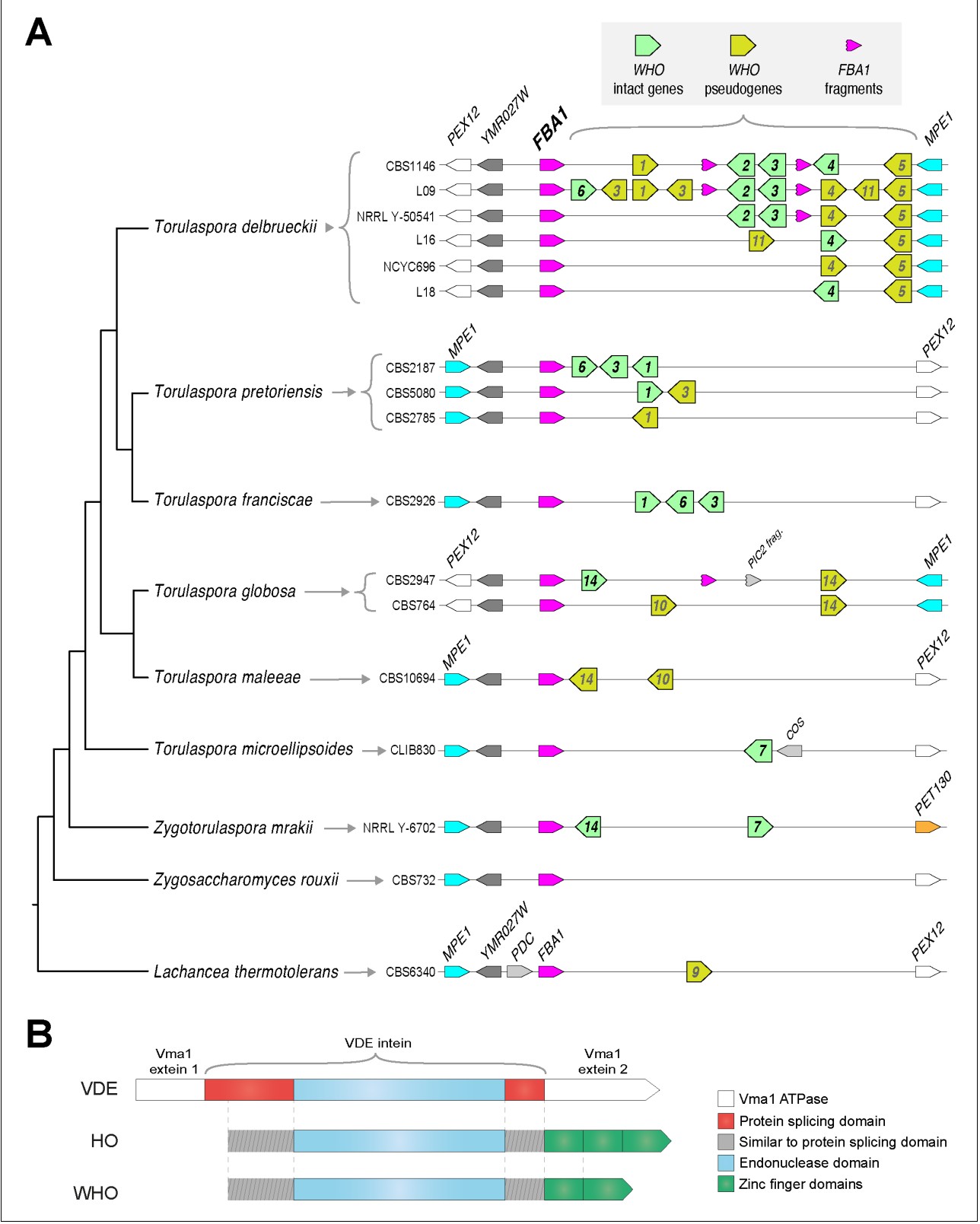

**Figure 1.** Genomic organization and domain structure of *WHO* genes. (**A**) Polymorphic clusters of *WHO* genes and pseudogenes downstream of *FBA1* in *Torulaspora* species. Multiple alleles are shown for *T. delbrueckii*, *T. pretoriensis*, and *T. globosa*. *WHO* genes are indicated by their family number. Fragments of the 3′ end of the *FBA1* gene are marked. Genomic views are schematic and not drawn to scale. The phylogenetic tree is based on

*Figure 1 continued on next page*

*Figure 1 continued*

*Shen et al. (2018)*. (B) Domain structure of HO, VDE and WHO proteins. The protein splicing domain is formed from two regions of the protein that flank the endonuclease domain (*Moure et al., 2002*).

The online version of this article includes the following figure supplement(s) for figure 1:

**Figure supplement 1.** Genomic organization and domain structure of *WHO* genes in *Lachancea* species.

fragments are about 400 bp long (full-length *FBA1* is 1080 bp). Some *FBA1* fragments contain frameshifts or internal stop codons, and others are intact.

The WHO proteins consist of an intein-like region followed by a zinc finger domain (*Figure 1B*). This structure is similar to HO, but distinct from VDE which has no zinc finger. WHO and HO are the only intein-zinc finger fusion proteins known in any organism. WHO and HO both have no exteins, and their intein domains both lack an amino acid motif that is normally found at the C-terminal end (motif G; *Pietrokovski, 1994*). Similar to HO, the zinc finger domains of the WHO proteins contain variable numbers (3-5) of a Cys-X-X-Cys motif. By BLAST searches, we found that the most similar zinc finger domains in other yeast proteins occur in orthologs of *S. cerevisiae* Ash1 (a regulator of *HO* transcription), which is an atypical type of GATA zinc finger domain (*Scazzocchio, 2000*; *Münchow et al., 2002*), though the level of sequence identity is low (maximally 38% over 76 residues).

## *WHO* gene/pseudogene clusters downstream of *FBA1* are common in the genera *Torulaspora* and *Lachancea*

*WHO* gene/pseudogene clusters are present downstream of *FBA1* in all five other species of the genus *Torulaspora* that we examined, and they again show within-species polymorphism in their gene content. We found three allelic *WHO* cluster arrangements among 9 *T. pretoriensis* strains, and two allelic arrangements in 2 *T. globosa* strains (*Figure 1A*). A pair of *WHO* genes is also present downstream of *FBA1* in *Zygotorulaspora mrakii*, which is closely related to *Torulaspora*. There are no *WHO* genes in the next most closely related genus, *Zygosaccharomyces*.

By database searches, we found that *WHO* genes also occur in the genus *Lachancea*, but they are absent from almost every other sequenced budding yeast genome. Eleven of the 12 sequenced *Lachancea* species have clusters of *WHO* genes or pseudogenes downstream of *FBA1*, and five of these species have intact *WHO* genes (*Figure 1—figure supplement 1*). The *WHO* clusters in four *Lachancea* species also contain duplicated fragments of the 3' end of *FBA1*. These structures in *Lachancea* are remarkably similar to the ones seen in *Torulaspora*, especially considering that these two genera are not closely related to each other, and there are no *WHO* genes in most other yeast genera. In addition, a few *Lachancea* genomes contain other *WHO* genes or pseudogenes at loci separate from *FBA1* (*Figure 1—figure supplement 1*).

In summary, *WHO* genes resemble homing endonuclease genes (*Gimble, 2000*). Like HO, but unlike any other homing endonucleases, the endonuclease domain is fused to a zinc finger domain. The structure of *WHO* genes, and their occurrence in clusters that are mixtures of intact genes and pseudogenes, suggests that they are part of a mobile genetic element.

## A WHO protein cleaves the *T. delbrueckii FBA1* gene in an allele-specific manner

We hypothesized that WHO proteins are homing endonucleases whose target is the *FBA1* gene. This hypothesis was motivated by two observations. First, *WHO* genes with very diverse sequences occur in tandem clusters downstream of *FBA1*, in both *Torulaspora* and *Lachancea*, suggestive of repeated integrations of different members of a mobile element family into the same target locus. Second, fragments of *FBA1* are present within the *WHO* gene clusters in both *Torulaspora* and *Lachancea* (*Figure 1* and *Figure 1—figure supplement 1*). All the *FBA1* fragments consist of only the 3' end of the gene, and many of them begin at approximately the same position (base 670–680 in the gene sequence), which we hypothesized could indicate a possible endonuclease cleavage site in *FBA1*.

To test the hypothesis that WHO proteins target the *FBA1* gene, we carried out experiments in *S. cerevisiae* because few tools exist for genetic manipulation of *Torulaspora* or *Lachancea*. We chose

*T. delbrueckii WHO6* for these experiments because it is intact, present in only a minority of the *T. delbrueckii* isolates we examined (3 of 15), and it is located at the end of the cluster closest to *FBA1* (*Figure 1A*). Together, these features suggested that *WHO6* could be the most recently-inserted *WHO* gene in the cluster in the strains that contain it, and therefore that *WHO6* is a good candidate for a homing element that is currently active and spreading through the *T. delbrueckii* population.

*FBA1* is an essential gene in most growth conditions (*Lobo, 1984*; *Schwelberger et al., 1989*; *Boles and Zimmermann, 1993*), so we reasoned that if it is the natural target of WHO endonuclease cleavage, then strains of *T. delbrueckii* that contain a *WHO6* gene should contain alleles of *FBA1* that are resistant to cleavage by Who6 endonuclease, whereas other *T. delbrueckii* strains might contain alleles that are sensitive to Who6. We constructed haploid strains of *S. cerevisiae* that contain the open reading frame (ORF) of *T. delbrueckii FBA1* (*TdFBA1*) integrated into the *ADE2* gene on chromosome XV. These strains also have the native *S. cerevisiae FBA1* gene on chromosome XI. We used two different alleles of *TdFBA1*: one from a *T. delbrueckii* isolate (strain L09) that has a *WHO6* gene downstream of it, and one from an isolate (CBS1146) that has no *WHO6* gene (*Figure 1A*). We then introduced a high copy-number panARS plasmid (pWHO6-HA) on which *WHO6* was expressed from the constitutive *T. delbrueckii TDH3* promoter (*Figure 2A*), with a 3xHA epitope tag at its 3' end. As described below, we found that this plasmid induces cleavage of the allele from strain CBS1146, but not of the allele from strain L09. Hence we designated the CBS1146 allele *TdFBA1-S* (for sensitivity to cleavage by Who6), and the L09 allele *TdFBA1-R* (for resistance). We also found that the plasmid does not induce cleavage of *S. cerevisiae FBA1*.

Our experiment is similar to one carried out by *Moore and Haber (1996)* who overexpressed HO so that it continually cleaved the *MAT* locus in haploid *S. cerevisiae* cells that had no *HML/HMR* loci. They found that the only cells that survived HO overexpression were ones in which inaccurate DNA repair ligated the chromosome back together but modified the target site sequence in such a way that HO could no longer cleave it, because chromosomes with accurate repairs were re-cleaved by HO. Similarly, in our experiment, the only haploid *S. cerevisiae* cells containing *TdFBA1-S* that survived overexpression of Who6 were ones in which the *TdFBA1-S* sequence became modified, either by gene conversion or by imprecise non-homologous end joining (NHEJ) (*Figure 2A*).

By genome sequencing, we found that in two independent experiments where pWHO6-HA was introduced into *S. cerevisiae* strains containing *TdFBA1-S*, this *T. delbrueckii FBA1* allele was modified by gene conversion with the native *S. cerevisiae FBA1* gene. In contrast, no gene conversion was seen when pWHO6-HA was introduced into *S. cerevisiae* strains containing the *TdFBA1-R* allele (two independent transformants), nor in control transformations in which a similar plasmid expressing 3xHA-tagged Green Fluorescent Protein (pGFP-HA) was transformed into *S. cerevisiae* strains containing *TdFBA1-S* or *TdFBA1-R*. Apart from the gene conversions at the *TdFBA1-S* transgene, no other nucleotide changes were detected in the genomes of the strains transformed with pWHO6-HA, and the native *S. cerevisiae FBA1* gene remained unchanged in this experiment.

To verify that the gene conversions were not caused by the 3xHA tag, we carried out additional experiments in which *S. cerevisiae* strains containing either *TdFBA1-S* or *TdFBA1-R* were each independently transformed 10 times with either pWHO6 (a plasmid expressing Who6 with no 3xHA tag) or pGFP-HA as a control. A single colony was chosen from each transformation, and its *ade2::TdFBA1* locus was amplified by PCR and sequenced. Approximately 100-fold fewer colonies were obtained from the transformations of pWHO6 into the *TdFBA1-S* strain than in any of the other three combinations of plasmid (pWHO6 or pGFP-HA) and allele (*TdFBA1-S* or *TdFBA1-R*). Evidence of cleavage of the *T. delbrueckii FBA1* ORF was observed in all 10 independent transformants of pWHO6 into the *TdFBA1-S* strain: five transformants showed gene conversion between *TdFBA1-S* and the native *S. cerevisiae FBA1* gene, and the other five had single-nucleotide insertions or deletions in *TdFBA1-S* consistent with cleavage and repair by imprecise NHEJ (*Figure 2B*). In contrast, no sequence changes at *ade2::TdFBA1* were detected in the 10 independent transformants of pWHO6 into the *TdFBA1-R* strain, nor (as expected) in any of the 20 transformants with pGFP-HA.

From these experiments we conclude that the *T. delbrueckii* isolate (L09) that contains the *WHO6* gene also contains, 588 bp upstream, an allele of *FBA1* (*TdFBA1-R*) that is resistant to cleavage by the Who6 endonuclease. It can therefore stably maintain this endonuclease gene in its genome. In contrast, an isolate (CBS1146) that has no *WHO6* gene contains an *FBA1* allele (*TdFBA1-S*) that is sensitive to cleavage by Who6.

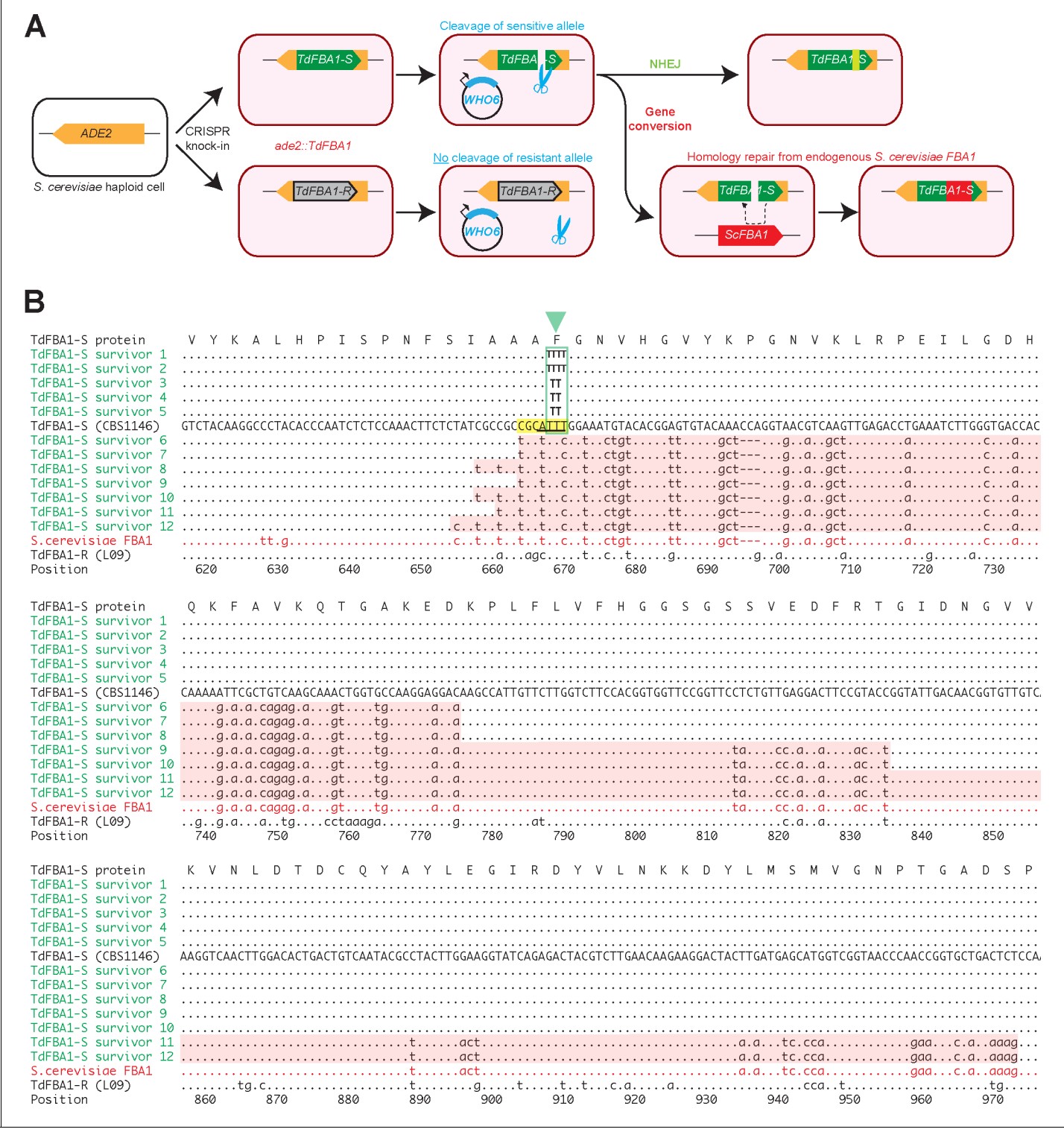

**Figure 2.** *WHO6* induces allele-specific DNA cleavage of the *T. delbrueckii FBA1* gene, with DNA repair by gene conversion or NHEJ. (**A**) Summary of the experiment. Haploid *S. cerevisiae* strains, containing a non-expressed *T. delbrueckii FBA1* ORF (*TdFBA1-S* or *TdFBA1-R* alleles) integrated at *ADE2*, were transformed multiple independent times with plasmids expressing WHO6 or WHO6-HA. In all transformations of strains containing the *TdFBA1-S* allele, the only colonies that survived expression of *WHO6* were ones in which *TdFBA1-S* underwent DNA cleavage and repair by gene conversion or imprecise NHEJ, changing its sequence and making it resistant to further cleavage. (**B**) Gene conversion and imprecise NHEJ events in *TdFBA1-S*. The reference DNA sequence (uppercase) shows the 3′ end of the *TdFBA1-S* allele from *T. delbrueckii* strain CBS1146. Survivors 1–5 are transformants in which *TdFBA1-S* was cleaved by Who6 and repaired by imprecise NHEJ near position 668 (green box and triangle; survivors 1 and 2 have a 1 bp

*Figure 2 continued on next page*

*Figure 2 continued*

insertion, and survivors 3–5 have a 1 bp deletion, relative to the sequence TTT in the reference). Survivors 6–12 are transformants in which *TdFBA1-S* was cleaved by Who6 and partially overwritten by gene conversion with the endogenous *S. cerevisiae FBA1* gene. Gene conversion regions are highlighted with pink backgrounds. The *TdFBA1-R* allele from *T. delbrueckii* strain L09, which is the natural host of *WHO6*, is also shown; this allele acquired no sequence changes among 10 independent pWHO6 transformants examined. A putative Who6 recognition site (yellow) and cleavage site with 4 bp 3' overhang (underlined) are marked. Survivors 10 and 11 are from transformations with pWHO6-HA; all other survivors are from transformations with pWHO6. The complete *TdFBA1* gene was sequenced from all transformants but only positions 616 to 975 are shown; there were no changes outside this region.

## The WHO endonuclease cleavage site in *FBA1*

Similar to the HO overexpression survivors in *Moore and Haber (1996)*, our transformants that contained the sensitive *TdFBA1-S* allele of *T. delbrueckii FBA1* and survived overexpression of Who6 have acquired mutations that can be inferred to have damaged the Who6 recognition or cleavage sites, making the cells resistant to Who6. These mutations enable us to identify the approximate location of the Who6 recognition and cleavage sites in *T. delbrueckii FBA1*.

The five transformants showing evidence of imprecise NHEJ (survivors 1–5 in *Figure 2B*) each sustained a single 1 bp insertion or deletion in the *TdFBA1-S* sequence, at the same site in the gene (positions 667–669: TTT→TTTT, or TTT→TT). Like other LAGLIDADG endonucleases, HO and VDE both make a staggered double-strand break with 4 bp 3' overhangs when they cleave DNA, and they have large (~24 bp) degenerate recognition sequences that span the cleavage site (*Nickoloff et al., 1990*; *Gimble and Thorner, 1993*; *Taylor et al., 2012*). We therefore infer that the overhang made by Who6 must include some or all of the TTT sequence centered on position 668 (*Figure 2B*). The HO recognition site at the *MAT* locus is moderately conserved between *S. cerevisiae* and other budding yeasts, and has at its core the sequence CGC<u>AACA</u>, where the 4 bp overhang is underlined. By analogy, we suggest that the core of the Who6 recognition site in the sensitive *TdFBA1-S* allele of *T. delbrueckii FBA1* is CGC<u>ATTT</u> (positions 663–669). In the resistant *TdFBA1-R* allele, 3 of these seven bases are different (Cag<u>cTTT</u>). In *S. cerevisiae FBA1*, which is also resistant to cleavage by Who6, 3 of 7 bases are different (tGC<u>tTTc</u>) (*Figure 2B*).

The seven transformants showing evidence of gene conversion (survivors 6–12, including two from pWHO6-HA transformations and five from pWHO6 transformations) each replaced a section of the *T. delbrueckii FBA1-S* sequence with the corresponding section of the *S. cerevisiae FBA1* gene from chromosome XI (*Figure 2B*). The *T. delbrueckii* and *S. cerevisiae* genes have 84% nucleotide sequence identity overall. The gene conversion tracts are asymmetrical: they extend rightwards (towards the stop codon of *FBA1*) from the cleavage site for 106–306 bp, whereas they extend leftwards for only 5–14 bp. This asymmetry suggests that one side of the WHO-induced double-strand break is more active in recombination than the other. Similarly, during mating-type switching in *S. cerevisiae*, the DNA on one side of the HO-induced break (the Z-side) participates in exchange with *HML/HMR*, whereas DNA on the Y-side remains inert until it is eventually clipped off (*Lee and Haber, 2015*).

In summary, the pattern of NHEJ events and gene conversions in *S. cerevisiae* cells that contain *TdFBA1-S* and survive continuous expression of a WHO protein is very similar to the pattern in cells that contain *MAT* and survive continuous expression of HO (*Moore and Haber, 1996*). We infer that WHO endonucleases cleave *FBA1* genes at approximately base 668, which is slightly upstream of the 5' ends of many of the *FBA1* fragments seen in *Torulaspora* and *Lachancea* species. The core of the putative recognition site of a WHO endonuclease is also similar to that of HO, with the sequence 5'-CGC-3' adjacent to the overhang.

## Repeated homing continually replaces the 3' end of *FBA1* and builds *WHO* clusters

The *TdFBA1-S* and *TdFBA1-R* alleles have only 85% nucleotide sequence identity downstream of base 668, which is remarkably low for two alleles from the same species. In contrast, they have 99% identity upstream of this position. More generally, among the full-length *FBA1* genes of the 15 *T. delbrueckii* isolates we analyzed, nucleotide sequence diversity is much higher in the 3' part of the gene (*Figure 3A*). Moreover, phylogenetic trees constructed from the 5' and 3' parts of *TdFBA1* (upstream and downstream of position 668) have contradictory topologies (*Figure 3B*). The

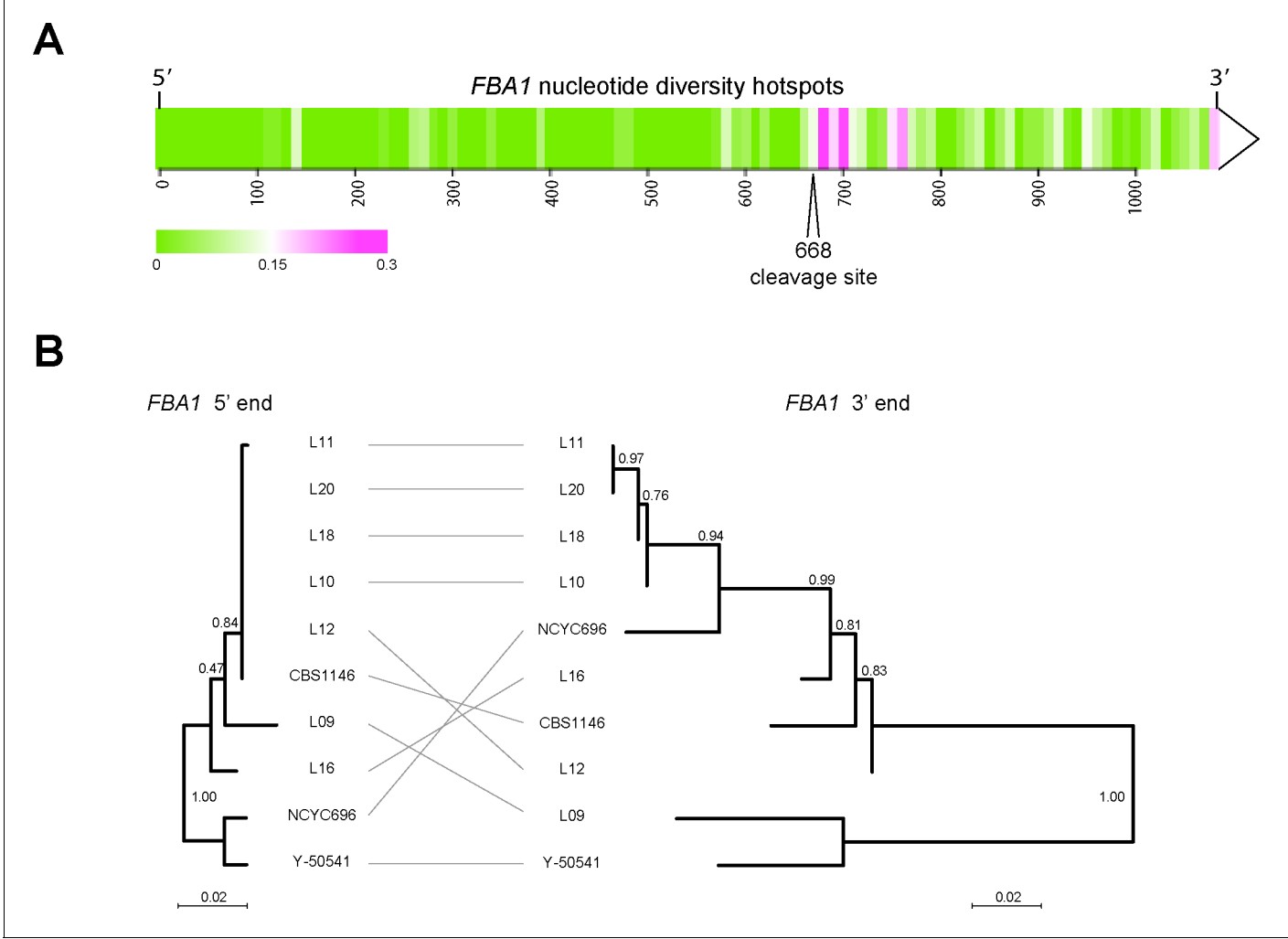

**Figure 3.** Different evolutionary dynamics of the 5' and 3' parts of *T. delbrueckii FBA1*. (**A**) Heatmap showing regions of nucleotide sequence diversity (π) among 15 sequenced *FBA1* alleles from *T. delbrueckii* isolates, plotted in 10 bp windows. (**B**) Inconsistency of phylogenetic trees obtained from the 5' and 3' ends of *FBA1* alleles from different *T. delbrueckii* strains (5' end: bases 1–667; 3' end: bases 669–1083). The alleles from strains CBS1146 and L09 are also called *TdFBA1-S* and *TdFBA1-R*, respectively. The trees are drawn to the same scale and were generated from nucleotide sequences using PhyML as implemented in Seaview v4.5.0 (*Gouy et al., 2010*) with default parameters. Bootstrap support from 1000 replicates is shown.

heterogeneous evolution of the two ends of *FBA1* in *T. delbrueckii*, and the presence of 3' *FBA1* fragments in its genome, suggest a mechanism for how the homing genetic element containing *WHO* genes operates.

We propose that *WHO* elements home into the *FBA1* locus by using a mechanism that involves replacing the 3' end of *FBA1*, thereby converting a sensitive *FBA1* allele into one that is resistant to the particular WHO protein encoded by the element (*Figure 4A*, left column). During meiosis in a heterozygous diploid cell, the WHO protein encoded by the donor allele cleaves the sensitive full-length *FBA1* gene of the recipient allele at position 668. The double-strand break is then repaired by using the *WHO*-containing chromosome as a template. The 3' region of the cleaved full-length *FBA1* gene interacts with the *FBA1* fragment in the template, resulting in incorporation of the *WHO* gene into the previously empty allele. After homing, the recipient chromosome contains a *WHO* gene located between a resistant full-length *FBA1* gene (a chimera of the recipient's previous 5' end and a copy of the donor's 3' end) and a new *FBA1* fragment formed from the recipient's previous 3' end. In this model, the *FBA1* fragment downstream of the donor's *WHO* gene is an essential part of the *WHO* element because it provides a region of homology that acts as a recombination site

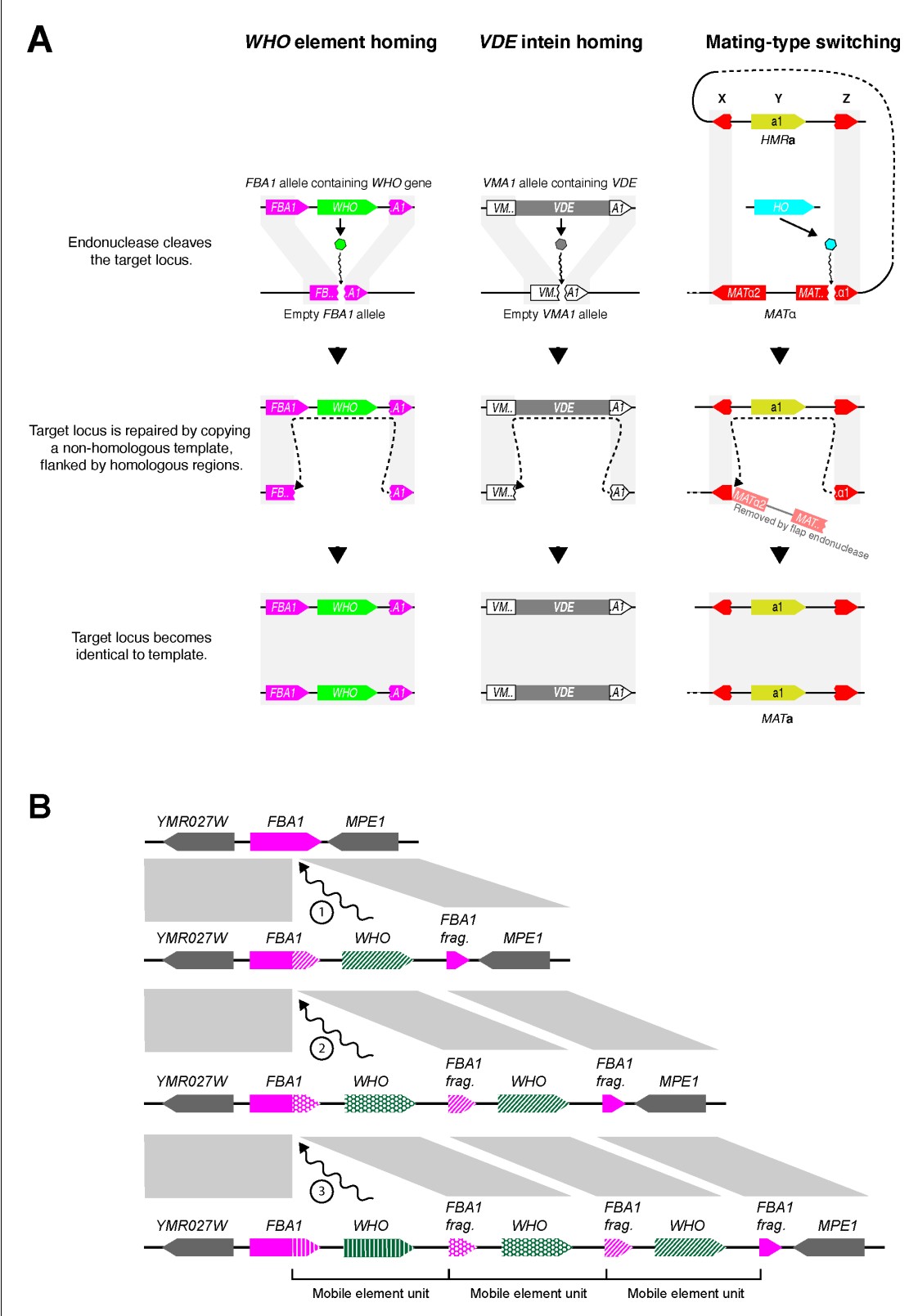

**Figure 4.** Proposed mechanism of *WHO* element homing. (**A**) Similarity of mechanisms of action of *WHO*, *VDE* and *HO*. The mechanism we propose for *WHO* elements integrating at *FBA1* is compared to the known mechanisms for *VDE* integrating into *VMA1* and for *HO*-mediated switching of the *MAT* locus (**Gimble and Thorner, 1992**; **Lee and Haber, 2015**). *WHO* and *VDE* homing occur between allelic chromosomes in a diploid cell, whereas mating-type switching occurs between *MAT* and *HML/HMR* loci in a haploid cell. Gray rectangles indicate regions of sequence identity. The column on

*Figure 4 continued on next page*

*Figure 4 continued*
the right shows mating-type switching from *MAT*α to *MAT*a in *S. cerevisiae*. Switching from *MAT*a to *MAT*α occurs by an identical mechanism; the core of the HO recognition site (CGCAACA) is the first 7 nucleotides of the Z region, which is present in both of the *MAT* alleles even though it is part of the *MAT*α1 gene sequence. The *HO* gene is on a different chromosome than *MAT-HML-HMR*. (B) Model for *WHO* cluster formation by successive integration of *WHO* elements. Every time a *WHO* element integrates into the locus, the 3' end of the full-length *FBA1* gene is replaced. The previous 3' end is pushed rightwards, together with any older *WHO* genes, after which they can decay into pseudogenes. The complete *WHO* mobile element unit consists of a *WHO* gene and the upstream 3' end of *FBA1*, which confers resistance to it.

(*Figure 4A*). The fragment is not part of the expressed *FBA1* gene so it does not need to maintain an open reading frame.

While the modified *FBA1* is now resistant to the newly acquired *WHO* gene, it may still be sensitive to other *WHO* genes. Repeated homing of multiple different *WHO* elements into the same chromosome will build tandem clusters of *WHO* genes and *FBA1* fragments, with the most recent elements being located closest to the full-length *FBA1* gene (*Figure 4B*). Each homing event replaces the 3' end of the full-length *FBA1* gene with the sequence from a different allele, causing its rapid evolution and discordant phylogenies. Over time, the *WHO* genes and *FBA1* fragments can decay into pseudogenes or become deleted, because they are not required for the aldolase function of *FBA1*. The functional unit of a *WHO* element can be defined as a *WHO* gene and the resistance-conferring *FBA1* 3' region upstream of it (*Figure 4B*).

In summary, *T. delbrueckii WHO* genes are part of a homing genetic element that targets *FBA1*. Our model for its mechanism of action explains how *WHO* clusters and *FBA1* fragments are formed, and the unusual chimeric mode of evolution of *FBA1*.

## Phylogeny of *WHO* genes shows an *FBA1*-associated backbone and multiple transpositions to other genomic loci

We investigated the phylogenetic relationship among *WHO*, *HO*, and *VDE* genes, using amino acid sequences inferred from intact genes and from some of the less-damaged *WHO* pseudogenes. In view of the high divergence among the sequences, the tree topology may not be fully accurate, but it permits identification of approximately 14 families of *WHO* genes (*Figure 5*). The *WHO* families form a monophyletic group, separate from *HO* and *VDE*. Most of the families are either *Torulaspora*-specific or *Lachancea*-specific, indicative of recent gene duplications within each genus. The *WHO2*, *WHO4*, *WHO5* and *WHO11* families are specific to *T. delbrueckii* so they must be young. Overall, the tree indicates a dynamic history of extensive *WHO* gene duplication and frequent formation of pseudogenes, consistent with the 'cycle of degeneration' expected for a homing genetic element (*Burt and Koufopanou, 2004*).

Although most *WHO* genes and pseudogenes are located downstream of *FBA1* genes (magenta branches in *Figure 5*), a few of them are not. It is striking that these non-*FBA1*-associated genes fall into a small number of clades (blue branches in *Figure 5*). The *WHO* genes in these clades seem to have lost their target specificity for *FBA1* and transposed to other places in the genome, and several of these *WHO* genes are intact. Most notably, the *WHO10* family includes five intact genes from *T. globosa* that are located at five different places in the genomes of the two isolates we sequenced (*Figure 6*). There is a *WHO10* pseudogene beside *FBA1* in one *T. globosa* isolate and the sister species *T. maleeae* (*Figure 1A*), indicating that *WHO10* was originally associated with *FBA1*. Another *WHO* clade unlinked to *FBA1* is *WHO8*, which is present only in two species of *Lachancea* (*Figure 5*).

We detected only a few *WHO* sequences in species other than *Torulaspora* (or *Zygotorulaspora*) and *Lachancea* in BLAST searches against the NCBI database, which includes genome sequences from hundreds of yeast species including members of almost every genus in the family Saccharomycetaceae (*Shen et al., 2018*). Thus the *WHO* family has a very limited phylogenetic distribution, and occurs mostly in two genera that are not sisters of each other (*Shen et al., 2018*). The few *WHO* sequences outside *Torulaspora* and *Lachancea* all lie in the *WHO13* and *WHO12* families, which are outgroups to the other *WHO* families (*Figure 5*), and most of them are pseudogenes. *WHO13* is *FBA1*-associated but *WHO12* is not. The *WHO13* sequences are all pseudogenes and were detected only in a small clade of *Kazachstania* species, downstream of *FBA1*. Interestingly, these species have an intron in *FBA1* at precisely the inferred WHO endonuclease cleavage site (*Figure 5—figure*

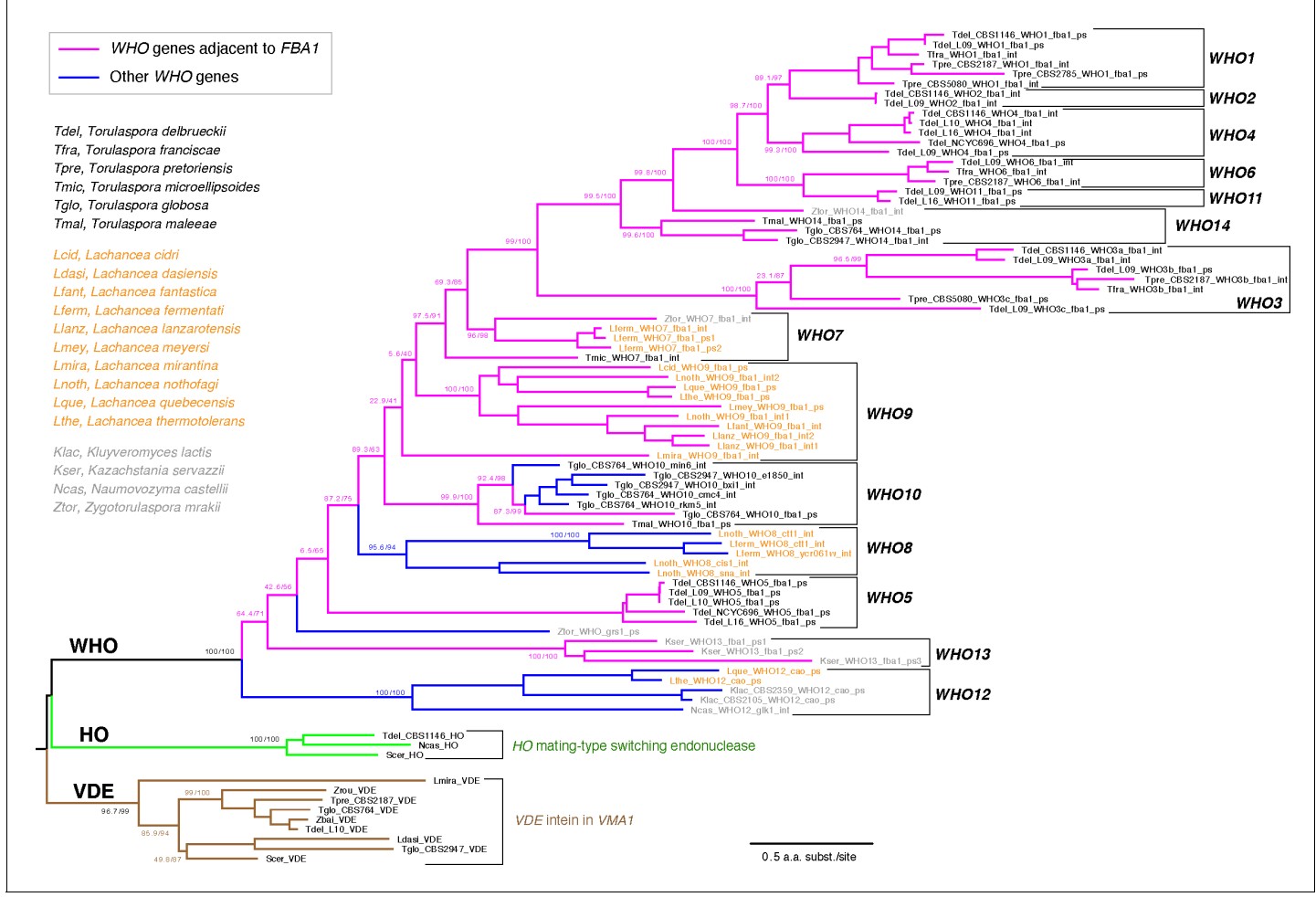

**Figure 5.** Families of *WHO* genes and their phylogenetic relationship to *HO* and *VDE*. Magenta branches indicate *WHO* genes that are located at the *FBA1* locus, and blue branches indicate *WHO* genes that are not beside *FBA1*. 14 *WHO* families are marked by brackets. Individual *WHO* gene names are colored by their source genus (black, *Torulaspora*; orange, *Lachancea*; gray, other genera). *WHO* gene names indicate the source species and strain number (if multiple strains were analyzed), *WHO* family (in uppercase), the name of a neighboring gene in the genome (in lowercase), and the suffix 'int' for intact *WHO* genes or 'ps' for *WHO* pseudogenes. Protein sequences were aligned using MUSCLE and filtered with Gblocks as implemented in Seaview v4.5.0 (***Gouy et al., 2010***). Badly degraded pseudogenes (relics) were not included. The tree was constructed by maximum likelihood using IQ-TREE v1.6.12 (***Trifinopoulos et al., 2016***), utilizing the built-in model finder option. Numbers on branches show support values from SH-aLRT and 1000 ultrafast bootstraps, separated by a slash (***Trifinopoulos et al., 2016***). The tree was rooted using VDE because WHO and HO share a zinc finger domain.

The online version of this article includes the following figure supplement(s) for figure 5:

**Figure supplement 1.** Some *Kazachstania* species have an intron in *FBA1* at a location corresponding to the *WHO* cleavage site in *Torulaspora*.

*supplement 1*), and this is the only intron in any budding yeast *FBA1* gene. In other eukaryotes, evolutionarily novel introns are gained at sites of double-strand DNA breakage (***Li et al., 2009***).

The position of *WHO12* as an outgroup to all the other *WHO* families (***Figure 5***) raises the possibility that the common ancestor of all the *WHO* families might have used a different gene as its original target, before changing target to *FBA1* after the *WHO12* family separated from the others. The only intact *WHO12* gene occurs in *Naumovozyma castellii*, where it is located beside *GLK1* (glucokinase). We sequenced the genomes of four strains of this species and found two with intact *WHO12*, and two with frameshifted *WHO12* pseudogenes. There was no structural polymorphism or high sequence divergence at this locus in *N. castellii*, and no *GLK1* gene fragments, so no evidence of active homing. The other *WHO12* sequences are pseudogenes in *K. lactis* (***Fabre et al., 2005***) and two *Lachancea* species (***Figure 1—figure supplement 1***), beside a *CAO* copper amine oxidase gene in all three cases. The *K. lactis WHO12* pseudogene lies between full-length *CAO* and a damaged

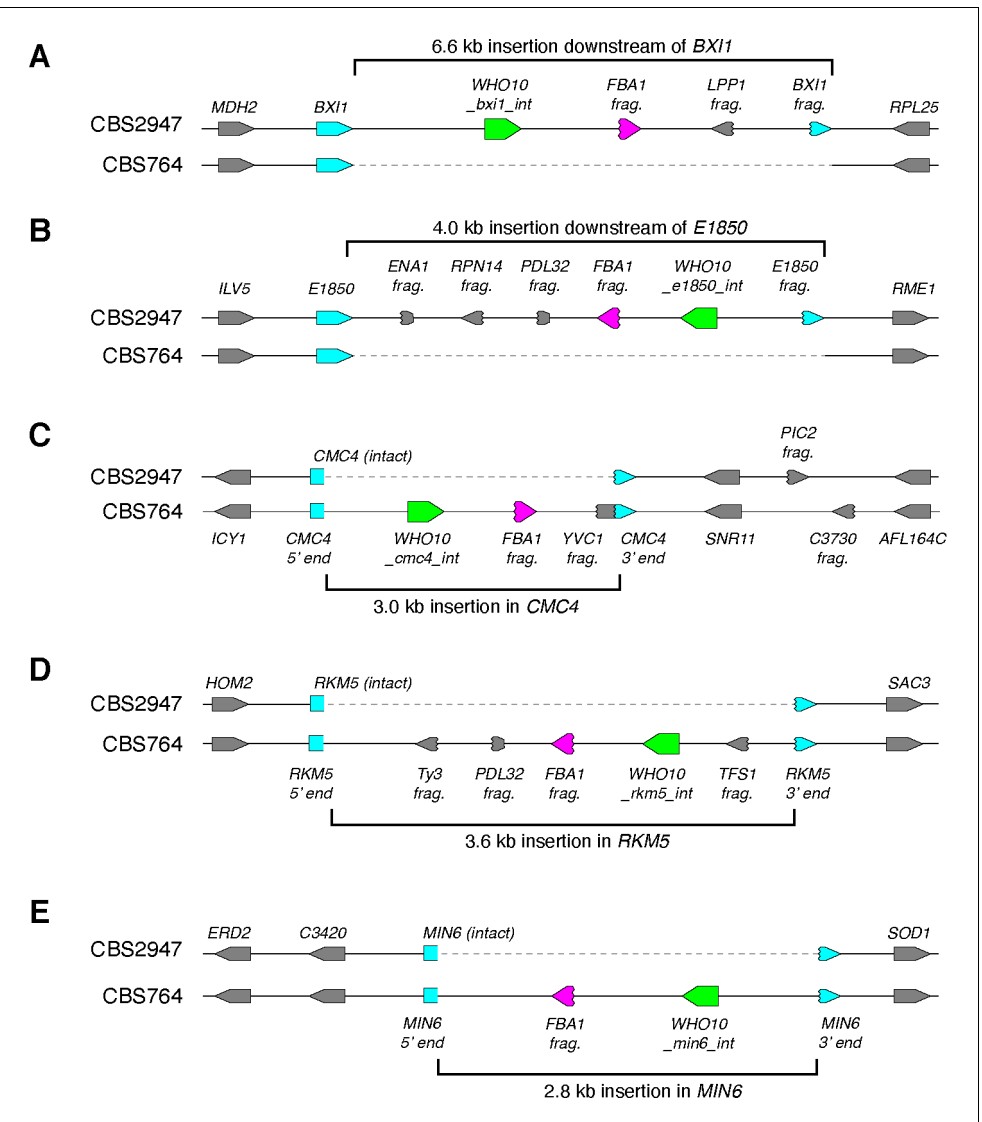

**Figure 6.** *WHO10* is an active mobile genetic element in *T. globosa* and integrates into loci other than *FBA1*. Each panel shows a pair of allelic regions from *T. globosa* strains CBS2947 and CBS764[T]. Intact *WHO10* genes are shown in green, *FBA1* fragments in magenta, and host genes in blue. (A-E) Five strain-specific insertions of *WHO10* elements into different host loci. At the *BXI1* and *E1850* loci of CBS2947 (A,B), the 3' end of the host gene became duplicated, whereas at the *CMC4*, *RKM5* and *MIN6* loci of CBS764 (C-E) the host gene was disrupted by the insertion and no part of it became duplicated. All the integrations contain a fragment of *FBA1* immediately downstream of *WHO10*, and most also contain fragments of other genes. Genes are named after their *S. cerevisiae* orthologs where possible. *E1850* is the *T. globosa* ortholog of *T. delbrueckii TDEL0E01850*, a gene with no homolog in baker's yeast.

fragment of *CAO*. It is therefore possible that *WHO12* is an active element targeting *CAO*, and that *CAO* pre-dates *FBA1* as the target of the whole *WHO* superfamily, but given the rarity of *WHO12* it seems more likely that *FBA1* is the ancestral target and *WHO12* is a family that lost its specificity for *FBA1* more recently, similar to *WHO10*.

In summary, the phylogenetic tree indicates that *WHO* genes have been located beside *FBA1* throughout most of their diversification. The *WHO* genes that are not now located beside *FBA1* are descended from *FBA1*-linked ancestors. The patchy taxonomic distribution of *WHO* genes suggests that they are native to the genus *Torulaspora* and/or *Lachancea* and have probably been transmitted between these genera by horizontal gene transfer. Horizontal transfer among budding yeast species

has previously been inferred for the *VDE* intein (**Koufopanou et al., 2002**; **Okuda et al., 2003**). The split between the zinc finger-containing proteins HO and WHO pre-dates the diversification of the *WHO* families.

## Recent transposition of *WHO10* genes in *Torulaspora globosa* confirms the mobility of *WHO* elements

The *WHO10* family has become amplified in *T. globosa*. In the two strains that we sequenced, strain-specific insertions of intact *WHO10* genes are present at five loci not linked to *FBA1* (**Figure 6**). The five Who10 proteins have only 72–80% amino acid sequence identity to one another. The *T. globosa WHO10* genes are located within regions of inserted DNA 2.8–6.6 kb long. In two cases, the inserted DNA includes a duplicated fragment of the 3' end of a host gene (*BXI1* and *E1850*) at one end, so that the host gene was not disrupted (**Figure 6A,B**). These duplications resemble the *FBA1* fragments seen downstream of *FBA1* in many species (**Figure 1A**). In the other three cases, the *WHO10*-containing insertion interrupts a host gene without forming a duplication, probably inactivating it (**Figure 6C,D,E**).

All five *T. globosa* insertions include an *FBA1* fragment immediately downstream of the *WHO10* gene, even though the insertions are not near the *FBA1* locus. Some insertions also contain fragments of various other genes, mostly from their 3' ends (**Figure 6**). The structure and variable location of the DNA insertions indicate that *WHO10* genes are part of a mobile genetic element that is active in *T. globosa*. The mobile element consists of *WHO10* and the gene fragments, which may be molecular fossils of previous host sites into which the element inserted.

The genomic evidence indicates that most *WHO* genes function as part of a homing genetic element that targets the *FBA1* locus (**Figure 4**). However, in *T. globosa* the *WHO10* family has lost its specificity for *FBA1* and become a more general mobile element rather than a homing element, resulting in the proliferation of intact *WHO10* genes to multiple other sites in the genome. How the *WHO10* genes become integrated into the non-*FBA1* sites in *T. globosa* is not clear, because there are no homologous flanking sequences to guide integration of *WHO10* into a double-strand break.

## Discussion

We have shown that *WHO* elements are homing genetic elements in the budding yeast genera *Torulaspora* and *Lachancea*, that primarily target the aldolase gene *FBA1*. They have diversified into a large family with very divergent endonuclease genes. Our model proposes that *WHO* elements home into sensitive alleles of *FBA1* by using a duplicated fragment of the 3' end of *FBA1* as a second region of homology downstream of the *WHO* gene (**Figure 4A**, left column). Homing replaces the 3' end of *FBA1*, making it resistant to cleavage by the element's WHO endonuclease. The DNA manipulation steps in *WHO*'s homing mechanism are identical to those that occur during *VDE* intein homing into *VMA1*, but the gene organization of *WHO* elements and their relationship to the host gene differ substantially from *VDE* (**Figure 4A**, first two columns). Resistance to endonuclease cleavage in *FBA1* comes from allelic sequence differences, whereas resistance in *VMA1* comes directly from interruption of the cleavage site by the *VDE* element (**Gimble and Thorner, 1992**).

*FBA1* can be described as the host gene for the *WHO* element, even though the element lies downstream of *FBA1* rather than interrupting it. This structural organization makes *WHO* elements different from the two currently recognized classes of homing genetic elements, which are inteins and intron-encoded homing endonucleases (**Belfort et al., 2005**; **Belfort, 2017**). In both of these other classes the homing element is a self-splicing entity, transcribed as an internal part of the host gene, that must be removed (by mRNA or protein splicing) in order to express the mature host protein. In contrast, *WHO* genes are transcribed independently of *FBA1* (some of them are in the opposite orientation to *FBA1*; **Figure 1A**), and *FBA1* will remain functional even if the *WHO* gene becomes a pseudogene. *WHO* elements therefore constitute a third structural class of homing element, and the only one with a propensity to form clusters. The mechanism of action of *WHO* elements has altered the evolutionary trajectory of their host gene *FBA1*, disrupting the normal vertical inheritance of this gene and leading to a chimeric mode of evolution in which the two ends of the gene have different histories (**Figure 3**).

The reason why *WHO* elements chose *FBA1* as their host gene is probably that aldolase is absolutely required for spore formation, due to its role in gluconeogenesis (**Dickinson and Williams,**

*1986*). Meiosis and sporulation require cells to be grown on a non-fermentable carbon source such as acetate, and in these conditions gluconeogenesis is necessary to make the glucose monomers used for synthesis of the polysaccharide layers of the spore wall, a late stage in the meiosis-sporulation pathway (*Neiman, 2005*; *Walther et al., 2014*). *S. cerevisiae fba1* mutants cannot make spores (*Lobo, 1984*; *Dickinson and Williams, 1986*) but they should not be blocked in meiosis, which is when *WHO* element homing is expected to occur. It is unlikely that the *FBA1* genes in either the donor or the recipient chromosome can be transcribed at the same time as DNA cleavage and recombination is occurring during homing. By temporarily inactivating *FBA1*, the *WHO* element may be able to delay the cell from progressing from meiosis into sporulation until homing has finished. Homing is likely to be a slow process, because mating-type switching takes more than an hour (*Lee and Haber, 2015*).

During mating-type switching (*Figure 4A*, right column), HO initiates a series of DNA manipulation steps that closely resemble the steps that occur during homing of *WHO* elements and the *VDE* intein. Together with the sequence similarity among the three proteins, this similarity of the molecular mechanisms indicates a shared evolutionary origin of the three processes. While the mechanisms of *WHO* and *VDE* homing are essentially identical, the mechanism of HO action at the *MAT* locus has diverged from them in two critical ways. First, the *HO* gene is not part of the template used for DNA repair. Second, switching occurs in haploids, whereas homing occurs in diploids during meiosis. There is no homologous chromosome for the cleaved *MAT* locus to interact with, so instead it interacts with *HML* or *HMR*.

Our results finally illuminate the origin of HO endonuclease. Based on the fact that WHO and HO share features that are otherwise unique – the presence of a zinc finger domain and the absence of exteins – we propose the following evolutionary model (*Figure 7*). (1) An intein from a bacterial source invaded the *VMA1* gene of an early budding yeast species to become *VDE*. (2) *VDE* subsequently duplicated and mis-homed into a zinc finger protein gene (possibly a paralog of *ASH1*), close to the 5′ end, to make a fusion gene that was the common ancestor of *WHO* and *HO*. The zinc finger directed the endonuclease to new target gene(s) in the genome. (3) The fusion gene became located between the target gene and a duplicated fragment of the target gene, forming a proto-*WHO* element. This step resembles some of the *WHO10* insertion sites seen in *T. globosa*. The target gene may have been *FBA1*, or possibly a different, unknown, gene. To function as a homing element, the proto-*WHO* element must have had a meiosis-specific promoter. (4) The proto-*WHO* element diversified and spread through yeast populations and into additional species, with *FBA1* as its main target. Occasional mis-homing events spread the element into new targets such as the *WHO10* locations in *T. globosa*. (5) At an early stage of diversification, a WHO endonuclease developed an ability to cleave *MATα1*, in a species that already contained a three-locus *MAT/HML/HMR* mating-type switching system, and became domesticated as *HO*. During domestication, the transcriptional regulation of the gene must have changed from meiosis-specific expression to haploid-specific expression, as well as gaining cell lineage and cell cycle constraints (*Stillman, 2013*). The boundary between the Y and Z regions of the *MAT* locus, which was previously variable among species, became permanently fixed at the site where the endonuclease cleaved *MATα1* (*Figure 4A*; *Hanson and Wolfe, 2017*).

Many examples are known of mobile genetic element genes that have been domesticated to take on a new role in the cell (*Volff, 2006*). In some of these examples, a domesticated endonuclease gene has retained its nucleolytic activity and functions in a programmed genome rearrangement process, such as the *RAG1* gene in V(D)J recombination in the immune system of jawed vertebrates (*Huang et al., 2016*), and the *PiggyMac* gene in elimination of germline sequences during development of the macronucleus in *Paramecium* (*Baudry et al., 2009*). In other examples the ability to cleave DNA has been lost, such as in the bacterial DUF199/WhiA family which originated as a LAGLI-DADG endonuclease but is now a regulator of transcription (*Kaiser et al., 2009*). HO has retained its endonuclease activity, and its origin from a homing element may help explain some unusual properties of this protein in vitro, such as its extreme catalytic inefficiency and its ability to attach to both ends of linear DNA molecules, forming loops visible by electron microscopy (*Jin et al., 1997*).

The domestication of a *WHO* element to become *HO* is similar to the domestication of the transposon-derived genes *KAT1* and *α3* to act as generators of double-strand breaks at the *MAT* locus during mating-type switching in *K. lactis* (*Barsoum et al., 2010*; *Rusche and Rine, 2010*; *Rajaei et al., 2014*). In all three cases, a mobile element gene was domesticated in a genome that

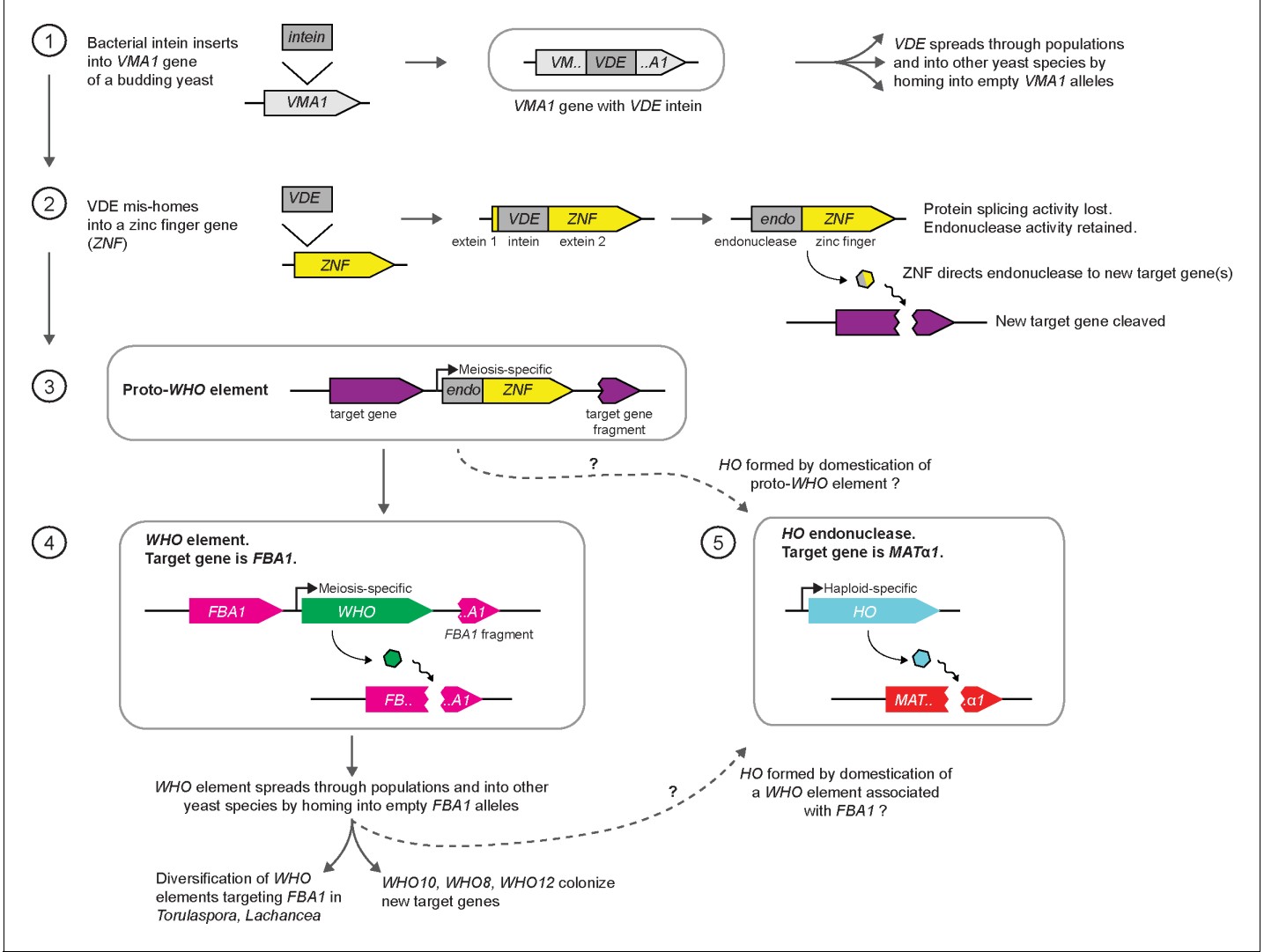

**Figure 7.** Model for the origin of *HO* by domestication of a *WHO* element. See Discussion for details. The dashed arrows indicate two possible routes to *HO*, from ancestral *WHO* elements that either were, or were not, specifically associated with *FBA1*.

already had a three-locus *MAT/HML/HMR* arrangement and probably switched mating types by a passive process based on homologous recombination without a specific mechanism for making a double-strand break at *MAT*. Why were mobile genetic elements repeatedly recruited into these switching systems? If we consider that, in any population of haploid cells, (1) mating-type switching can only increase a cell's probability of mating, (2) mating leads to the formation of a diploid and inevitably to meiosis, even if many vegetative generations later, and (3) homing genetic elements can only home during meiosis, it logically follows that it is in a homing genetic element's self-interest to increase the frequency of mating-type switching (*Hanson and Wolfe, 2017*). Thus, a *WHO* element could increase its rate of spread into empty *FBA1* alleles in a population, if its WHO protein developed a secondary activity of cleaving the *MAT* locus as well as cleaving *FBA1*. Importantly, haplontic species such as *Torulaspora* and *Lachancea* require a nutritional signal to mate, so mating-type switching is unlikely to be followed by immediate mating of switched cells with their clonal relatives as occurs in *S. cerevisiae*. Instead, we expect that a switched cell in a haplontic species could go through many cycles of mitotic division before mating, which increases the opportunity for switched cell lineages to disperse and outbreed, and therefore increases the opportunity for homing elements to spread. Since frequent and accurate switching are probably favored by natural selection

(*Hanson et al., 2014*), the subsequent steps that domesticated a *WHO* element to form the non-mobile and exquisitely regulated gene *HO* (*Stillman, 2013*) would also have been advantageous.

## Materials and methods

**Key resources table**

| Reagent type (species) or resource | Designation | Source or reference | Identifiers | Additional information |
|---|---|---|---|---|
| Gene (*Torulaspora delbrueckii*) | *FBA1* | *Gordon et al., 2011* | *TDEL0B06660*; NCBI: XM_003679958 | Strain CBS1146; *TdFBA1-S* allele |
| Strain, strain background (*Saccharomyces cerevisiae*) | IMX585 | *Mans et al., 2015* | | *S. cerevisiae* strain with integrated *CAS9* gene |
| Strain, strain background (*Torulaspora delbrueckii*) | L09; L10; L11; L12; L13; L15; L16; L18; L19; L20 | This paper | | Strain collection of Lallemand, Inc |
| Strain, strain background (*Torulaspora delbrueckii*) | NCYC696 | National Collection of Yeast Cultures (UK) | | |
| Strain, strain background (multiple *Torulaspora* species) | All CBS strains | Westerdijk Fungal Biodiversity Institute (Netherlands) | | |
| Strain, strain background (*Torulaspora pretoriensis*) | UWOPS 83–1046.2 | MA Lachance, University of Western Ontario (Canada) | | |
| Strain, strain background (*Zygotorulaspora mrakii*) | NRRL Y-6702 | CP Kurtzman, USDA Agricultural Research Service | | |
| Strain, strain background (*Naumovozyma castellii*) | Y056 (NRRL Y-12630$^T$); Y174 (CBS4310); Y287 (CBS3006); Y668 (CBS1579) | Jure Piškur, Lund University (Sweden) | | See *Spírek et al., 2003* |
| Recombinant DNA reagent | pMEL13 plasmid | *Mans et al., 2015* | | |
| Sequence-based reagent | *ADE2.Y* sgRNA template | *DiCarlo et al., 2013* | | |
| Recombinant DNA reagent | pIL75-*KanMX* plasmid | *Liachko and Dunham, 2014* | | |
| Recombinant DNA reagent | pWHO6 | This paper | | DNA sequence is in *Supplementary file 3* |
| Recombinant DNA reagent | pWHO6-HA | This paper | | DNA sequence is in *Supplementary file 3* |
| Recombinant DNA reagent | pGFP-HA | This paper | | DNA sequence is in *Supplementary file 3* |
| Sequence-based reagent | *Sc ade2::TdFBA1-S* repair template | This paper | | DNA sequence is in *Supplementary file 3* |
| Sequence-based reagent | *Sc ade2::TdFBA1-R* repair template | This paper | | DNA sequence is in *Supplementary file 3* |
| Sequence-based reagent | Primers for PCR of *Sc ade2::TdFBA1* locus | This paper | | 5'-TGACCACGTT AATGGCTCC-3' and 5'-CACCAGCTCCA GCGATAATTG-3' |
| Software, algorithm | SPAdes assembler | *Bankevich et al., 2012* | V3.11.1 | RRID:SCR_000131 |
| Software, algorithm | HGAP3 assembler | *Chin et al., 2013* | | |

## Yeast isolates and genome sequences

Genomes analyzed in this study are listed in *Supplementary files 1* and *2*. New genome sequences were obtained as follows. *T. delbrueckii* strains L09–L20 were from the strain collection of Lallemand Inc (Montréal, Canada), generously provided by Dr. Caroline Wilde. They were sequenced at the University of Missouri core facility (Illumina, SE 1 × 50 bp). *T. delbrueckii* strain NCYC696 data were downloaded from opendata.ifr.ac.uk/NCYC on 23-Feb-2017 as unassembled Illumina sequence reads (PE 2 × 100 bp). *T. globosa* strains CBS764[T] and CBS2947 were purchased from the Westerdijk Institute (Netherlands) and sequenced using both Illumina (PE 2 × 150 bp; BGI Tech Solutions, Hong Kong) and Pacific Biosciences Sequel technologies (1 SMRT cell; Earlham Institute, Norwich, UK). *T. pretoriensis* strain CBS2187[T] (Illumina, PE 2 × 100 bp + 6 kb library MP 2 × 100 bp) and *T. franciscae* strain CBS2926[T] (Illumina, PE 2 × 100 bp) were sequenced and assembled at INRAE Montpellier. Eight other *T. pretoriensis* strains (CBS2785, CBS5080, CBS9333, CBS11100, CBS11121, CBS11123, CBS11124 from the Westerdijk Institute, and UWOPS 83–1046.2 from M.A. Lachance, University of Western Ontario) were sequenced at the Earlham Institute using their proprietary LITE protocol for the Illumina platform. *Zygotorulaspora mrakii* strain NRRL Y-6702[T] was obtained from the USDA Agricultural Research Service (Peoria, IL, USA) and sequenced at the Earlham Institute using both Pacific Biosciences RSII (4 SMRT cells) and Illumina LITE methods. *Naumovozyma castellii* strains Y056, Y174, Y287 and Y668 were gifts from Prof. Jure Piškur (Lund University, Sweden) and were sequenced at the Earlham Institute using the Illumina LITE method.

Cultures were grown under standard rich-medium conditions. DNA for Illumina sequencing was harvested from stationary-phase cultures by homogenization with glass beads followed by phenol-chloroform extraction and ethanol precipitation. Purified DNA was concentrated with the Genomic DNA Clean and Concentrator-10 (Zymo Research, catalog D4010). DNA for PacBio sequencing was prepared as in *Ortiz-Merino et al. (2017)*.

Illumina data were assembled using SPAdes version v3.11.1 (*Bankevich et al., 2012*). PacBio data were assembled using HGAP3 (*Chin et al., 2013*).

Other genome sequences used in this study were taken from the NCBI database. The previously published genome sequences for *Torulaspora* species are from *Gordon et al. (2011)*, *Gomez-Angulo et al. (2015)*, *Tondini et al., 2018*, *Galeote et al. (2018)* and *Shen et al. (2018)*; *Lachancea* species are from *Souciet et al. (2009)*, *Sarilar et al. (2015)*, *Vakirlis et al. (2016)*, *Freel et al. (2016)* and *Kellis et al. (2004)*; and *Kluyveromyces* species are from *Dujon et al. (2004)* and *Varela et al. (2019)*.

## Construction of *S. cerevisiae ade2::TdFBA1* strains

*S. cerevisiae* strains in which the coding region (ORF) of *T. delbrueckii FBA1* was integrated into the *S. cerevisiae ADE2* gene, in opposite orientation to *ADE2* so that it is not functional, were constructed using CRISPR-Cas9 as follows. The *ADE2*-targeting sgRNA *ADE2*.Y from *DiCarlo et al. (2013)* was synthesized as a gene fragment by Integrated DNA Technologies, and inserted into the sgRNA plasmid pMEL13 from *Mans et al. (2015)* by restriction digestion and ligation. This plasmid was then transformed into *S. cerevisiae* strain IMX585 expressing Cas9 (*Mans et al., 2015*), together with a repair template containing the *T. delbrueckii FBA1* ORF (*TdFBA1-S* or *TdFBA1-R* allele) flanked with homology to *S. cerevisiae ADE2* in reverse orientation (bases 564456..564832 and 565952..566366 of *S. cerevisiae* chromosome XV). Sequences of the *ade2::TdFBA1-S* and *ade2:: TdFBA1-R* constructs are given in *Supplementary file 3*. Transformants were selected on YPAD (YPD (Formedium) supplemented with 40 μg/ml adenine sulfate (Sigma)) containing 200 μg/ml G418. *ADE2* knockouts were identified by formation of red colonies. Successful integrants were confirmed by PCR amplification of the *ade2::TdFBA1* locus and Sanger sequencing (Eurofins). Two replicate *ade2::TdFBA1-S* strains were designated C1 and C4, and two replicate *ade2::TdFBA1-R* strains were designated L1 and L3.

## *WHO6* expression plasmid construction

Replicating plasmids constitutively expressing Who6, Who6-HA, or GFP-HA were constructed in the panARS replicating vector pIL75 (*Liachko and Dunham, 2014*) containing a *KanMX* marker. The nucleotide sequences of these plasmids (pWHO6, pWHO6-HA, pGFP-HA) are given in

*Supplementary file 3*. In pWHO6, the *WHO6* gene from *T. delbrueckii* strain L09 was placed under the control of the promoter and terminator of the *T. delbrueckii* glyceraldehyde-3-phosphate dehydrogenase gene *TDH3* (*TDEL0E04750*). These regions were amplified by PCR from *T. delbrueckii* genomic DNA using high fidelity polymerase (New England Biolabs, M0492S) and inserted into pIL75 by restriction digestion and ligation. Plasmid pWHO6-HA is identical to pWHO6 except that its *WHO6* gene is fused to a C-terminal 3xHA tag. A similar control plasmid (pGFP-HA) was made containing a GFP gene fused to 3xHA tag (GBlock made by Integrated DNA Technologies and inserted into pIL75 by restriction digestion and ligation), under the control of the *T. delbrueckii* *TDH3* promoter and terminator.

## Gene conversion assays

The *S. cerevisiae ade2::TdFBA1* strains (C1, C4, L1, L3) as described above were transformed with the plasmids pWHO6, pWHO6-HA, or pGFP-HA. Multiple independent transformations were made to ensure that all gene conversion or NHEJ events recovered were independent. First, pWHO6-HA and pGFP-HA were each transformed into strains C1, C4, L1 and L3, and the whole genomes of these eight strains were sequenced, resulting in the unexpected discovery of gene conversion at the *ade2::TdFBA1* locus when C1 and C4 were transformed with pWHO6-HA (called survivors 11 and 10, respectively, in *Figure 2B*). Genome sequencing was done by BGI Tech Solutions using a BGI-SEQ instrument with 50 bp single-end reads. Second, pWHO6 (no 3xHA tag) and pGFP-HA were each transformed 10 times into strains C4 and L1. The *ade2::TdFBA1* locus from each transformant was amplified by PCR with a high-fidelity polymerase (New England Biolabs, M0492S) and Sanger sequenced. Primers for amplification were TGACCACGTTAATGGCTCC and CACCAGCTCCAGCGATAATTG.

For transformation, *S. cerevisiae* cells were grown overnight in liquid cultures of YPAD. Cultures were reinoculated in 50 ml and grown to mid-log phase. Cells were incubated with 1M LiAc, 50% PEG, salmon sperm DNA and plasmid DNA for 30 min, and then heat shocked at 42°C for 15 min. Transformants were selected on YPAD containing 200 µg/ml G418. Only one colony was picked from each transformation plate. The number of colonies obtained on the plates with the combination C4 + pWHO6 was dramatically lower (approximately 100-fold) than on the three other combinations. Individual colonies were grown in liquid YPAD overnight, before genomic DNA was harvested using a QIAamp DNA Mini kit (Qiagen).

## Acknowledgements

We thank Caroline Wilde, Weilong Hao, Warren Albertin and Robert Mans for strains, Devin Scannell and Mike Eisen for preliminary data on *T. globosa*, and Raúl Ortiz-Merino and Amanda Lohan for assistance. We thank Sara Hanson, Geraldine Butler and Wolfe lab members for comments on the manuscript. We thank Jo Dicks for permission to use the NCYC696 sequence data, which were produced by the UK National Collection of Yeast Cultures, in partnership with The Earlham Institute, using funding awarded to the Institute of Food Research, Norwich, UK by the Biotechnology and Biological Sciences Research Council. This study was supported by Science Foundation Ireland (13/IA/1910) and the European Research Council (789341).

## Additional information

### Funding

| Funder | Grant reference number | Author |
| --- | --- | --- |
| Science Foundation Ireland | 13/IA/1910 | Kenneth H Wolfe |
| European Research Council | 789341 | Kenneth H Wolfe |

The funders had no role in study design, data collection and interpretation, or the decision to submit the work for publication.

## Author contributions
Aisling Y Coughlan, Conceptualization, Formal analysis, Investigation, Writing - original draft, Writing - review and editing; Lisa Lombardi, Investigation, Writing - review and editing; Stephanie Braun-Galleani, Alexandre AR Martos, Virginie Galeote, Frédéric Bigey, Sylvie Dequin, Investigation; Kevin P Byrne, Data curation, Investigation, Writing - review and editing; Kenneth H Wolfe, Conceptualization, Formal analysis, Writing - original draft, Writing - review and editing

## Author ORCIDs
Kenneth H Wolfe (iD) https://orcid.org/0000-0003-4992-4979

## Decision letter and Author response
Decision letter https://doi.org/10.7554/eLife.55336.sa1
Author response https://doi.org/10.7554/eLife.55336.sa2

# Additional files

## Supplementary files
• Supplementary file 1. *T. delbrueckii* genome sequence data used in this study.

• Supplementary file 2. Other genome sequence data used in this study.

• Supplementary file 3. Sequences of the plasmids pWHO6, pWHO6-HA and pGFP-HA, and of the *ade2::TdFBA1-S* and *ade2::TdFBA1-R* constructs.

• Transparent reporting form

## Data availability
Key nucleotide sequence data is provided in Supplementary File 3. New genome sequences have been deposited at NCBI. Their Bioproject numbers are in the dataset table and also in Supplementary files 1 and 2.

The following datasets were generated:

| Author(s) | Year | Dataset title | Dataset URL | Database and Identifier |
|---|---|---|---|---|
| Coughlan AY, Lombardi L, Braun-Galleani S, Martos AAR, Galeote V, Bigey F, Dequin S, Byrne KP, Wolfe KH | 2020 | Torulaspora franciscae genome sequencing | https://www.ncbi.nlm.nih.gov/bioproject/PRJNA622240 | NCBI BioProject, PRJNA622240 |
| Coughlan AY, Lombardi L, Braun-Galleani S, Martos AAR, Galeote V, Bigey F, Dequin S, Byrne KP, Wolfe KH | 2020 | Torulaspora delbrueckii genome sequencing | https://www.ncbi.nlm.nih.gov/bioproject/?term=PRJNA623898 | NCBI BioProject, PRJNA623898 |
| Coughlan AY, Lombardi L, Braun-Galleani S, Martos AAR, Galeote V, Bigey F, Dequin S, Byrne KP, Wolfe KH | 2020 | Torulaspora delbrueckii strain: NCYC696 Genome sequencing | https://www.ncbi.nlm.nih.gov/bioproject/?term=PRJNA623891 | NCBI BioProject, PRJNA623891 |
| Coughlan AY, Lombardi L, Braun-Galleani S, Martos AAR, Galeote V, Bigey F, Dequin S, Byrne KP, Wolfe KH | 2020 | Torulaspora pretoriensis genome sequencing | https://www.ncbi.nlm.nih.gov/bioproject/?term=PRJNA623867 | NCBI BioProject, PRJNA623867 |

| Coughlan AY, Lombardi L, Braun-Galleani S, Martos AAR, Galeote V, Bigey F, Dequin S, Byrne KP, Wolfe KH | 2020 | Torulaspora globosa CBS764 genome sequencing and assembly | https://www.ncbi.nlm.nih.gov/bioproject/?term=PRJNA625704 | NCBI BioProject, PRJNA625704 |
| --- | --- | --- | --- | --- |
| Coughlan AY, Lombardi L, Braun-Galleani S, Martos AAR, Galeote V, Bigey F, Dequin S, Byrne KP, Wolfe KH | 2020 | Torulaspora globosa CBS2947 genome sequencing and assembly | https://www.ncbi.nlm.nih.gov/bioproject/?term=PRJNA625705 | NCBI BioProject, PRJNA625705 |
| Coughlan AY, Lombardi L, Braun-Galleani S, Martos AAR, Galeote V, Bigey F, Dequin S, Byrne KP, Wolfe KH | 2020 | Zygotorulaspora mrakii strain:NRRL Y-6702 Genome sequencing and assembly | https://www.ncbi.nlm.nih.gov/bioproject/?term=PRJNA625702 | NCBI BioProject, PRJNA625702 |
| Coughlan AY, Lombardi L, Braun-Galleani S, Martos AAR, Galeote V, Bigey F, Dequin S, Byrne KP, Wolfe KH | 2020 | Naumovozyma castellii genome sequencing | https://www.ncbi.nlm.nih.gov/bioproject/?term=PRJNA623732 | NCBI BioProject, PRJNA623732 |

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
