## [Decision Letter]

Thank you for submitting your article "The yeast mating-type switching endonuclease HO is a domesticated member of an unorthodox homing genetic element family" for consideration by *eLife*. Your article has been reviewed by three peer reviewers, one of whom is a member of our Board of Reviewing Editors, and the evaluation has been overseen by Patricia Wittkopp as the Senior Editor. The following individual involved in review of your submission has agreed to reveal their identity: Laura Rusche (Reviewer #2).

The reviewers have discussed the reviews with one another and the Reviewing Editor has drafted this decision to help you prepare a revised submission.

Summary:

In this work the authors sought to explain the evolutionary history of a well-studied gene – the endonuclease HO – that has been integral in our understanding of DNA break repair and mating-type switching. In doing so, they discovered a new kind of homing genetic element present in a few species of fungi – the WHO elements – that are phylogenetically and structurally distinct from other homing systems. These WHO elements primarily proliferate into sensitive, allelic sites (like other homing elements) but they apparently can also be copied to other non-allelic sites in genomes. A clever experiment was conducted in *S. cerevisiae* to demonstrate that a heterologously expressed Td WHO element cleaves a specific site in its proposed target gene, TdFBA1. This experiment also demonstrated that the double-strand break can be repaired by non-homologous end-joining or by homologous recombination. In the latter case, the conversion track extends primarily in one direction from the cut site. These properties, combined with the arrangement of WHO clusters downstream of *FBA1* genes, allowed the authors to propose a model for the spread of WHO elements into new *FBA1* alleles following mating of a WHO+ and a WHO- strain. This represents a new class of homing element, where the endonuclease is separable from the site of cleavage/conversion, with implications for our understanding of genome organization and evolution. The findings of this study extend our understanding of how transposable elements become domesticated for cellular purposes and offer a solution to the mystery of the source of the yeast HO gene.

The reviewers had two main comments that they would like the authors to add to the manuscript.

Essential revisions:

1) Could the authors further discuss the broad applicability of their findings? For example, the finding of an unusual chimeric mode of evolution for the *FBA1* gene appears to have repercussions for gene and genome evolution and challenges some of our basic assumptions about the vertical inheritance of genes. Similarly, it seems the findings of this study extend our understanding of how transposable elements become domesticated for cellular purposes.

2) It seems (at least) a missing step in the model is the evolution of the HO promoter from the WHO promoter. Did the well-studied cell-cycle regulation and Ash1 repression of the HO evolve from the WHO ancestor? This is also related more generally to the mechanism for WHO replication, because unlike the intein ancestor, WHOs cannot rely on the host gene promoter. In the authors model, did this promoter came from the ancestral ZNF gene? Was this ancestral gene was Ash1? The reviewers would like some discussion about the (evolution of) expression of the WHO and HO genes.

Suggestion:

The authors propose a specific sequence in *FBA1* as the target of WHO cleavage. To nail down this prediction, the authors could mutate the sequence in the sensitive *FBA1* allele and show that it becomes resistant. This is a straight-forward experiment and would strengthen the conclusion. However, the reviewers did not believe this is required for the conclusions of the study, and therefore they leave it as a suggestion to the authors.

---

## [Author Response]

Essential revisions:1) Could the authors further discuss the broad applicability of their findings? For example, the finding of an unusual chimeric mode of evolution for the FBA1 gene appears to have repercussions for gene and genome evolution and challenges some of our basic assumptions about the vertical inheritance of genes. Similarly, it seems the findings of this study extend our understanding of how transposable elements become domesticated for cellular purposes.

We have expanded the Discussion section to include more discussion of these points. We now comment on the chimeric mode of evolution of *FBA1* in Discussion paragraph two, and we put HO into a broader context of domesticated mobile element genes in Discussion paragraph five.

2) It seems (at least) a missing step in the model is the evolution of the HO promoter from the WHO promoter. Did the well-studied cell-cycle regulation and Ash1 repression of the HO evolve from the WHO ancestor? This is also related more generally to the mechanism for WHO replication, because unlike the intein ancestor, WHOs cannot rely on the host gene promoter. In the authors model, did this promoter came from the ancestral ZNF gene? Was this ancestral gene was Ash1? The reviewers would like some discussion about the (evolution of) expression of the WHO and HO genes.

We have amended our evolutionary model (Figure 7) and the associated text (Discussion paragraph four and five) to include a suggestion of changes in the promoter, to convert it from meiosis-specific expression in diploids (for *WHO*) to expression in haploids in the G1 phase of the cell cycle (for *HO*). We do not want to speculate too much on this topic because we do not have any experimental data about how *WHO* gene transcription is regulated. The ancestral zinc finger gene in our model cannot have been *ASH1* itself, because *ASH1* is intact in both *Torulaspora* and *Lachancea*, but it may have been a paralog of *ASH1* and we added this in paragraph four of the Discussion.

Suggestion:The authors propose a specific sequence in FBA1 as the target of WHO cleavage. To nail down this prediction, the authors could mutate the sequence in the sensitive FBA1 allele and show that it becomes resistant. This is a straight-forward experiment and would strengthen the conclusion. However, the reviewers did not believe this is required for the conclusions of the study, and therefore they leave it as a suggestion to the authors.

Our laboratory is closed due to the COVID-19 pandemic, so we are not able to do any experiments at the moment and we must respectfully decline the suggestion. However, we agree that investigating the cleavage site specificities of WHO proteins would be an interesting avenue for future research.